# The Effectiveness and Toxicity of Frameless CyberKnife Based Radiosurgery for Parkinson’s Disease—Phase II Study

**DOI:** 10.3390/biomedicines11020288

**Published:** 2023-01-20

**Authors:** Bartłomiej Goc, Agata Roch-Zniszczoł, Dawid Larysz, Łukasz Zarudzki, Małgorzata Stąpór-Fudzińska, Agnieszka Rożek, Grzegorz Woźniak, Magdalena Boczarska-Jedynak, Leszek Miszczyk, Aleksandra Napieralska

**Affiliations:** 1Radiotherapy Department, MSC National Research Institute of Oncology Gliwice Branch, 44-102 Gliwice, Poland; 2Department of Head and Neck Surgery for Children and Adolescents, University of Warmia and Mazury in Olsztyn, 10-561 Olsztyn, Poland; 3Department of Radiology and Diagnostic Imaging, MSC National Research Institute of Oncology Gliwice Branch, 44-102 Gliwice, Poland; 4Department of Radiotherapy Planning, MSC National Research Institute of Oncology Gliwice Branch, 44-102 Gliwice, Poland; 5“Kangur” Centre for Treatment of CNS Disorders and Child Development Support in Katowice, 40-594 Katowice, Poland; 6Neurology and Restorative Medicine Department, Health Institute dr Boczarska-Jedynak, 32-600 Oświęcim, Poland

**Keywords:** Parkinson’s disease, radiosurgery, frameless radiosurgery

## Abstract

Frame-based stereotactic radiosurgery (SRS) has an established role in the treatment of tremor in patients with Parkinson’s disease (PD). The low numbers of studies of frameless approaches led to our prospective phase 2 open-label single-arm clinical trial (NCT02406105), which aimed to evaluate the safety and efficacy of CyberKnife frameless SRS. Twenty-three PD patients were irradiated on the area of the thalamic ventral nuclei complex with gradually increasing doses of 70 to 105 Gy delivered in a single fraction. After SRS, patients were monitored for tremor severity and the toxicity of the treatment. Both subjective improvement and dose-dependent efficacy were analysed using standard statistical tests. The median follow-up was 23 months, and one patient died after COVID-19 infection. Another two patients were lost from follow-up. Hyper-response resulting in vascular toxicity and neurologic complications was observed in two patients irradiated with doses of 95 and 100 Gy, respectively. A reduction in tremor severity was observed in fifteen patients, and six experienced stagnation. A constant response during the whole follow-up was observed in 67% patients. A longer median response time was achieved in patients irradiated with doses equal to or less than 85 Gy. Only two patients declared no improvement after SRS. The efficacy of frameless SRS is high and could improve tremor control in a majority of patients. The complication rate is low, especially when doses below 90 Gy are applied. Frameless SRS could be offered as an alternative for patients ineligible for deep brain stimulation; however, studies regarding optimal dose are required.

## 1. Introduction

Parkinson’s disease (PD) is the second most common neurodegenerative disease in adults [1]. The majority of patients with PD present with tremor, which can significantly limit patients’ quality of life [1,2,3]. The standard treatment according to NICE Guidelines is pharmacotherapy with levodopa [4,5]. Unfortunately, tremor is often not sufficiently controlled using oral medication, and different treatment modalities need to be employed to mitigate this symptom. Deep brain stimulation (DBS) is a therapy of choice and good outcomes of such intervention have been reported [4,5,6]. However, it is an invasive neurosurgical procedure, and some patients are not candidates for such intervention [2,4]. The alternatives for these patients include lesion surgery, magnetic resonance-guided stereotactic radiofrequency thalamotomy and focused ultrasound, with comparable results having been reported [2,4,5,6,7,8]. GammaKnife (GK) stereotactic radiosurgery (SRS) was also found to be a useful tool in decreasing severity of tremor [4,7,9,10,11,12,13,14,15,16].

SRS can be implemented on a GK-based system or on a LINAC-based system, such as CyberKnife (CK). So far, the vast majority of reports have considered the use of frame-based GK systems, leaving a place for the assessment of similar, frameless procedures (we found only one report based on CK SRS) [14,17,18]. Taking advantage of years of experience in implementing CK-based SRS in our department, we designed this clinical trial to assess the efficacy of decreasing tremor level achieved by SRS targeted to thalamic nuclei complex. An analysis of the dose that could be safely delivered and effectively used in this clinical scenario was performed.

## 2. Materials and Methods

We conducted a prospective phase 2 open-label single-arm clinical trial (NCT02406105), protocol available at: https://clinicaltrials.gov/ct2/show/NCT02406105. The trial protocol was approved by the institutional review board and the ethics committee on 13 August 2014 (approval number KB/430-40/14) and performed according to the Helsinki Declaration. The study was designed to initially include 27 patients, with the primary endpoint being safety of the procedure. Toxicity was evaluated using RTOG/EORTC Acute Radiation Morbidity Scoring Criteria and Late Radiation Morbidity Scoring Schema [19]. Secondary outcomes included tremor reduction evaluation using the Global Impression of Change Scale (GICS) and patients’ self-assessment of treatment effect. According to the GICS, values of +4 to +2 were regarded as improvement, from +1 to −1 as stagnation and from −2 to −4 as progression of the tremor. The analysis of changes in patients’ quality of life, cognition and mental health were also conducted with the use of the Parkinson’s Disease Questionnaire 39, the Mini-Mental State Exam and the Beck Depression Inventory. This article provides the primary endpoint analysis and the outcomes regarding changes in neurologic functions. 

### 2.1. Inclusion Criteria

Between January 2015 and March 2020 we recruited adult patients with significant tremor, secondary to PD diagnosis. The main inclusion criteria were idiopathic PD, based on the UK Parkinson’s Disease Society Brain Bank Diagnostic Criteria, lack of effective pharmacotherapy, contraindications to DBS procedure or refusal to undergo such treatment, and informed consent for participation in the study and for SRS. All of the patients were referred for consultation with an experienced neurologist—movement disorders specialist. Patients were required to have an Eastern Cooperative Oncology Group (ECOG) performance status (PS) of no worse than 2.

All patients were obliged to have head magnetic resonance imaging (MRI), basic laboratory tests, an electrocardiogram and a blood pressure check-up. Before SRS planning procedures, all of the patients had neurological, neuropsychological and speech pathology assessment. Patients’ ability to understand proposed treatment modality and cooperation with the team was mandatory.

### 2.2. Exclusion Criteria

The main exclusion criteria were the following: age under 18 years old, pregnancy, other than PD induced tremor, concurrent dementia (Mini Mental Score > 24) or psychosis. In addition, patients in poor PS or with atrophic cerebral changes or structural changes in basal nuclei were excluded from the study. Prior brain radiotherapy was not allowed. All patients were required to be on a stable medication schema for at least a one-month period prior to CK SRS.

### 2.3. Radiosurgery

SRS was planned on the basis of the fusion of high-quality (1 mm thick slices) contrast-enhancement computed tomography (CT) and MRI (1 mm thick slices, T1 and MPR sequences were required). The structures used for fusion included enhanced blood vessels and internal brain structures. To maintain the highest level of reproducibility, all of the procedures were performed using a personalised vacuum mattress, a thermoplastic head mask and, optionally, in an attempt to minimize the influence of tremor on potential motion of the patient during treatment, Elastic band embracing the patient’s shoulders and chest. All patients were in a supine position with arms along the body. They were positioned with their head in a symmetric position in the sagittal and longitudinal axis (a line connecting outer eye corner and external auditory canal was perpendicular to the table’s surface). 

Target contouring was performed by the main investigator (LM) with the aim of an experienced neuroradiologist. The clinical target volume (CTV) in all patients was a complex of the thalamic nuclei: ventralis oralis anterior (VoA) and ventralis oralis posterior (VoP), contralaterally to the dominant symptom side. The CTV was defined geometrically in the following steps: identification of anterior and posterior commissure, creating a line connecting both of the structures, creating a second line from the middle of the above (midcommisural point) 12 mm laterally. The lateral end of the second line was the lower edge of the target, whereas all targets were cylindrical volume with base diameter of 1–2 mm and length of 3 mm cranially. The exact measurements were based on a stereotactic atlas and protocol applied in Italy [20,21,22]. No margin was added. The dose constraints for organs at risk were adapted from Timmermann’s study (see Table 1) [23,24].

In each case, SRS planning was performed using a Multiplan system dedicated to CK treatment. The dose was prescribed as 100% in the central point in target volume and whole target volume was covered with 80% isodose. The total dose ranged from 70 to 105 Gy delivered in single fraction. The dose was escalated by 5 Gy every 3 patients after at least 3 months of observation for toxicity of the procedure.

The procedure for each patient required a short hospital stay for close monitoring post-procedure. Premedication included hydroxyzine and, optionally, midazolam. The SRS was performed in dedicated immobilization and the patient’s position was continuously verified using a 6D skull-tracking system. The planned treatment time was less than one hour. The anti-oedematous treatment was started immediately after SRS.

### 2.4. Follow-Up

According to the trial’s protocol, control visits were planned every 3 months for the first year after SRS. Then, the visits were supposed to occur at longer intervals, approximately every 6 months. A comprehensive neurologic exam with treatment toxicity evaluation was performed at each follow-up visit. MRI imaging scans were also performed at each visit.

### 2.5. Statistical Analysis

Median follow-up was calculated using the Kaplan–Meier method with the reversed meaning of status indicator. The date of the visit on which examination revealed or the patient reported more pronounced severity of the tremor was regarded as the date of disease progression. Overall survival was calculated using the Kaplan–Meier method.

## 3. Results

### 3.1. Patient and Treatment Data

Due to unexpected toxicity observed in two of the patients and the tragic death of the main investigator, only 24 out of 27 planned patients were included into the study. From this number, only twenty-three received SRS and two more were lost from follow-up after the treatment (one who received 95 Gy and one who received 105 Gy). Thus, this report presents the results of 21 patients. Detailed characteristics of the study group are presented in Table 2.

Dose constraints were met, and treatment planning details and doses delivered to the nearby organ at risk (OAR) are presented in Table 3. Patients were treated with multiple 6 X MV beams, and the mean number of beams was 136 (range 109–170). Median treatment duration was 44 min (range 38–52).

### 3.2. Follow-Up and Treatment Response

Median follow-up was 23 months. During the study period, one patient died due to COVID-19 infection. Two-year overall survival was 94%.

Each patient’s PS and tremor were examined and evaluated by the treatment team during each of the follow-up visits. Treatment results of the whole group are presented in Table 4. Details of the treatment response in individual patients are presented in Appendix A. Two patients did not respond to SRS; the tremor severity of these patients increased compared to pre-SRS visits, and they did not declare any improvement as a result of SRS. All others responded to SRS with at least stagnation of the tremor severity. Reduced tremor severity was observed in 15 patients, and 6 experienced stagnation. The median time to reach maximum treatment response was 6 months (range 3–35). The majority of patients had treatment response observed early, usually during the first follow-up visit. However, four patients also experienced late treatment response, with the decline in tremor severity being observed 12 to 14 months after SRS. What is more, in three cases a very late response was observed—decline of tremor severity was observed after 24 (2 cases) and 35 months (1 case). Constant improvement, evaluated as decreased or no progression of tremor severity during the whole follow-up, was obtained in 14 patients (67%). Among the responders, worsening of tremor severity during the follow-up was observed in five patients (24%). The median time to progression of the tremor was 6 months (range 3–36 month). No improvement after SRS was declared by eight patients (38%) during the last follow-up visit; however, only two did not observe any benefit from SRS with regard to tremor severity during the whole follow-up (nonresponders).

The median total dose used in our study was 85 Gy, and comparison of the low-dose (≤85 Gy) and high-dose (>85 Gy) groups showed that exactly the same number of patients responded to the treatment (67%). Lack of response was observed in each of the arms in one patient. Looking at the length of the treatment response relative to the total dose applied, we observed that longer response was dominating in the low dose group. The median length of tremor reduction was 24.5 months versus 9 months in the low- and high-dose groups, respectively.

### 3.3. Treatment Toxicity

Two patients experienced vascular toxicity which was probably related to the treatment applied. Both of them showed changes in magnetic resonance imaging within the irradiation field, and both experienced progression of neurologic deficits and hemiparesis which did not resolve fully after applied treatment. One of them was irradiated with a total dose of 95 Gy, and one with 100 Gy.

## 4. Discussion

Trigeminal neuralgia, temporal lobe epilepsy, essential tremor and PD are functional disorders in which SRS is suitable for patients refractory to surgery or medication or who cannot tolerate invasive procedures [4,7,13,25]. DBS of the subthalamic nuclei is a surgical procedure commonly offered to PD patients with advanced motor complications related to dopaminergic replacement and providing motor benefit since the 1980s [2,4,7,13,21,25,26]. Permanent brain damage has rarely been observed in patients who have qualified for such a procedure; however, cognitive decline, affective disorders, hypophonia and apathy have been reported [20]. The results of lesioning by radiofrequency ablation or by GK SRS are similar to those induced by DBS. What is important, compared to DBS, is that both of them are less expensive, and both are noninvasive methods and carry lower risk of infection [2,7,21,25,26,27]. However, since such treatment is irreversible, safety concerns are an important aspect of lesion therapy [2,18].

The experience of frameless SRS in some PD patients has been studied; however, the literature in this area is sparse [14,17,18]. The high precision of delivery, shortening of delivery time, use of the head and neck thermoplastic mask and excellent results associated with frameless SRS have led to further studies in patients with functional disorders. Fusion of MRI with CT imaging is associated with very good visualization of anatomic landmarks, which allows for precise and safe SRS treatment delivery [25,27,28,29]. It is worth noting that a few studies involving GK SRS showed that some distortions in the position of the frame could be observed, while frameless SRS could be delivered with less than 1 mm uncertainty [28,29,30,31].

Several targets for SRS have been investigated [14,17,20,21,22]. The subthalamic nucleus is the most commonly targeted lesion, followed by the nucleus ventralis intermedius, the globus pallidus and the putamen [16,17,21]. Current International Stereotactic Radiosurgery Society guidelines identify the ventral intermediate nucleus as the most optimal target [16]. What remains the most important factor for success of each of the procedures is correct target identification, which will make the image-guided lesion therapy more standard. Treatment planning with the aim of MRI, CT and ventriculography allows for visualization and identification of target structures [20,21,22]. In our study, the target was delineated on the basis of a stereotactic atlas and protocol applied in Italy [14,20,21,22]. SRS was performed in dedicated immobilization, and the patient’s position was continuously verified using a 6D skull-tracking system.

Until now, the largest group of PD patients treated with frameless SRS was studied by Khattab et al. [18]. In their prospective single-arm trial, 33 patients with essential tremor (23 patients) or PD (10 patients) were included. A total dose of SRS was 156–160 Gy delivered in one fraction. Improvement in all evaluated aspects regarding tremor, handwriting and functional disability was noted. The overall treatment response rate was 83% at 6 months, and quality of life improved by 57% [18]. Franzini et al. described CK SRS thalamotomy of two PD patients who received 70 Gy (the first one) and 90 Gy (the second one), with a very good response and a major reduction in tremor being noted [14]. Kim et al. described a case report of a PD patient who received 140 Gy targeted on the thalamic ventral intermediate nucleus with a very good treatment response. Tremor decreased after 10 months and completely resolved after another year of observation [17]. Results observed in our group are comparable to those reported by other authors. Nevertheless, a wide range of total doses was applied, and based on the response rate and length of tremor reduction observed in our group and in the study by Franzini et al., it is possible that dose escalation is not needed to provide a satisfactory treatment outcome [11,14]. Lack of serious complications in low-dose groups also favours such a choice; however, doses in GK SRS studies are much higher. Current International Stereotactic Radiosurgery Society guidelines advocate the use of 130 Gy, but those guidelines refer only to GK SRS [16].

It is hard to compare the results of frameless and frame-based SRS, since experience with GK thalamotomy is much longer and more than 1500 patients received such lesion therapy. However, based on recent studies, similar response rates were described [11]. So far, three prospective studies involving GK SRS of patients with essential and Parkinsonian tremor have been published with no randomized clinical trial [10,32,33]. Witjas et al. reported tremor reduction in 54% of 50 treated patients. The ventral intermediate nucleus was the treatment target, and a single shot of 130 Gy was prescribed [32]. A similar target and treatment were applied in a study by Ohye et al.; however, here, a much higher response rate of 81% was observed [10]. In contrast to those reports, a prospective blinded study by Lim S-Y et al. reported much worse outcomes for GK thalamotomy. It is worth noting that, out of 14 evaluated patients, only two showed significant tremor reduction, though significant benefits in activities of daily living post-SRS were demonstrated [33]. Other retrospective studies describing GK SRS thalamotomy showed that this treatment provides good results (a response rate of between 56% and 100%, with an acceptable rate of side effects) in those who cannot undergo DBS [2,7,9,11,12,13,22,26].

Reported tolerance of frameless SRS shows that this treatment is safe and well tolerated [18]. Khattab et al. observed two cases of Grade 2 headaches which resolved with medication (anti-inflammatory drugs and dexamethasone), which corresponds with a 6% rate of complications [18]. No complications after CK SRS were reported in the Franzini et al. study of two patients; however, some radiological changes (small regions of necrosis and oedema) in the treatment area were observed, but with no related clinical symptoms [14]. Moreover, in the study by Kim et al., no complications were observed [17]. Frame-based SRS can produce complications related to cranial fixation, such as pain, bleeding with subcutaneous or subgaleal hematoma, infection or sinus fractures, which could be avoided in a frameless approach [14,17,18]. Prospective studies of GK SRS show that the number of serious complications is relatively low [10,32,33]. Witjas et al. described only one case of transient hemiparesis and excessive oedema around the thalamotomy target [32]. No permanent clinical complications were observed in the study by Ohye et al. [10]. However, Lim S-Y reported serious adverse neurologic complications in 3 out of 18 patients treated with GK [31]. A similar hyper-response to GK thalamotomy was reported in several other studies. The rate of serious complications which did not resolve with time ranges from single cases to 8.4% of the patients [2,7,9,10,11,12,13,16,22,25,26,27,32,33]. Factors hypothesized to be connected with the risk of hyper-response are target location, overall medical condition and fragility of the patient, but so far in the majority of cases the development of symptomatic radiation effects is considered to be idiosyncratic [26]. Across all the studies, the most common side effects described were motor problems that ranged from mild weakness to hemiparesis and dysphagia [2,7,9,10,11,12,13,16,22,26,27,32,33]. This corresponds with the rate and type of complications observed in our study, with two cases of hyper-response and haemorrhage in the site of the lesion, resulting in permanent hemiparesis of the patients.

There are several limitation to our study. In addition to the prospective nature of the study, it is an unblinded observational and nonrandomized trial with no control group or comparisons of the effect made before and after SRS. The small size of the group, the type of irradiation and the doses used made it hard to compare with other reports. What is more, generalizability of our findings is limited by our study design and the short follow-up in some patients. There were several missing data, due to dropout after baseline, which could be considered as significant; however, we made an attempt to contact all of these patients and gather missing information. It is worth noting that this is the largest study of PD patients who received frameless SRS to be published so far.

## 5. Conclusions

Frameless CyberKnife-based SRS of patients with Parkinsonian tremor is a well-tolerated treatment and could improve tremor control in majority of patients who are not candidates for invasive procedures. The longest response was observed in the group treated with doses of between 70 and 85 Gy. Prospective randomized studies are needed to better evaluate the response and toxicity rate in this group.

## Figures and Tables

**Table 1 biomedicines-11-00288-t001:** Dose constraints for nearby organs at risk.

Organs at Risk (OAR)	OAR’s Dose Constraints [Gy]
Capsula interna	Dmax 35
Brainstem	Dmax 8 Max point dose 15 V10 < 0.5 cc
Chiasm	Dmax 8 Max point dose 1
Optic nerves	Dmax 8
Lenses	Dmax 4

cc—cubic centimetre, D—dose, Gy—gray, max—maximum, Vx—volume which received dose of x Gy.

**Table 2 biomedicines-11-00288-t002:** Characteristics of the study group.

Patient Characteristics	Value (%)	
Age at the time of radiosurgery	Median 64 years (range 53–81, SD ± 6.6)	
Gender	Female Male	3 (14%) 18 (86%)
Performance status (ECOG)	0 1	7 (33%) 14 (67%)
Tremor dominant side	Right Left	14 (67%) 7 (33%)
Duration of tremor (months)	Median 96 (range 36–192)	
Parkinson’s disease medications	Levodopa Benserazide Carbidopa Amantadine Rasagiline Biperiden Entacapone Other DOPA receptor agonists Other anti-tremor drugs Other drugs (mainly antidepressants)	19 (90%) 15 (71%) 8 (38%) 4 (19%) 1 (5%) 2 (10%) 2 (10%) 4 (19%) 6 (29%) 8 (38%)
Daily levodopa dose (milligrams)	Median 800 (range 0–1800)	
Comorbidities	Lack of comorbidities Cardiovascular diseases Osteoarthritis Diabetes mellitus Type 2 Cancer (in anamnesis) No data	4 (19%) 9 (43%) 6 (29%) 6 (29%) 6 (29%) 1 (5%)

**Table 3 biomedicines-11-00288-t003:** Doses delivered to the target volume and nearby organs at risk.

Volume	Dose Delivered (Gray)	
Clinical target volume	Median	85 (range 70–105)
Capsula interna	Maximum dose Mean dose	8.09–34.01 (median 23.79) 2.35–10.10 (median 4.76)
Brainstem	Maximum dose Mean dose	2.51–10.31 (median 8.00) 0.53–1.34 (median 0.84)
Chiasm	Maximum dose	0.65–8.49 (median 1.70)
Optic nerves	Maximum dose	0.19–4.08 (median 0.60)
Lenses	Maximum dose	0.01–0.24 (median 0.20)

**Table 4 biomedicines-11-00288-t004:** Treatment results of the whole group.

	Time of Control Visit	3 Months	6 Months	9 Months	12 Months	Last Follow-Up Visit
	Number of Patients at Follow-Up Visit	19	16	14	16	21
**Tremor evaluation**						
Less severe		12 (57%)	3 (14%)	1 (5%)	3 (14%)	4 (19%)
No changes		6 (29%)	10 (48%)	11 (52%)	11 (52%)	14 (67%)
More severe		1 (5%)	3 (14%)	2 (10%)	2 (10%)	3 (14%)
No data		2 (9%)	5 (24%)	7 (33%)	5 (24%)	-
**Subjective improvement**						
Yes		16 (76%)	11 (52%)	8 (38%)	11 (52%)	13 (62%)
No		3 (14%)	5 (24%)	6 (29%)	5 (24%)	8 (38%)
No data		2 (9%)	5 (24%)	7 (33%)	5 (24%)	-

## Data Availability

Data available upon request.

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
