# Peer review of "The Effectiveness and Toxicity of Frameless CyberKnife Based Radiosurgery for Parkinson’s Disease—Phase II Study"

_biomedicines, 2023, doi:10.3390/biomedicines11020288_

Round 1
Reviewer 1 Report
Review of a manuscript “An Effectiveness and Toxicity of CyberKnife Based Radiosurgery for Parkinson Disease – phase II study” by Bartłomiej Goc and coauthors
Parkinson’s disease is a frequent and severe neurodegenerative disorder, for which here is no efficient treatment modifying the course of the disorder. One of the typical traits of Parkinson’s disease is tremor. The authors designed a clinical trial to assess the efficacy of reducing tremor level achieved by Frame-based radiosurgery (SRS) targeted to thalamic nuclei complex. The implementation of the trial is based on a long-term authors experience in using this method for Parkinson’s disease patients. This is an important area of biomedical science and the results presented in the manuscript will be interesting for the readers of the journal. The manuscript is scientifically valid, technically accurate and ethically sound
The following corrections and additions should be made.
Overall. The authors should be consistent in writing Parkinson’s disease or Parkinson disease. In the Abstract they use Parkinson disease, in the introduction – Parkinson’s disease.
Abstract
Line 26. The abbreviated form for “Frame-based radiosurgery” (SRS) may be confusing because it does not correspond to the first letters of the term. Stereotactic radiosurgery (SRS) may be easier for understanding.
Line 27. “Very scare experience” it is uncertain what the authors wanted to say by this. They should explain it more clearly.
Line 32-33:“The median follow-up was 23 months and one patient died (Covid-19 infection)” The sentence should be rewritten as follows: ”The median follow-up was 23 months and one patient died after Covid-19 infection”.
Introduction
Line 44-45. After the sentence “Parkinson`s Disease (PD) is the second most common neurodegenerative disease in adults.” The authors should add the citation to a recent review on Parkinson’s disease: ”Biomarkers in Parkinson’s Disease”. Chapter in a book Peplow P.V., Martinez B., Gennarelli T.A. (eds) Neurodegenerative Diseases Biomarkers. 2022. Neuromethods, vol 173. pp 155-180. Humana, New York, NY. https://link.springer.com/protocol/10.1007/978-1-0716-1712-0_7
Materials and Methods
“Due to unexpected toxicity observed in two of the patients” Can the authors be more specific and describe what type of toxicity it was?
Discussion
Line 239: ”What remains the most important for success of each of the procedure, is correct target identification, thus image guided lesion therapy became standard” the sentence should be rewritten as follows:” What remains the most important for success of each of the procedure, is correct target identification, which will make the image guided lesion therapy more standard”
Line 246: “Until now, the largest group of PD patients treated with frameless SRS was reported by Khattab et al.
The authors should add a citation for Khattab et al. publication. The same concerns the reference on line 280.
Author Response
We would like to thank the editor and the reviewers for careful and thorough reading of this manuscript and for the thoughtful comments and constructive suggestions, which help to improve the quality of this manuscript. Our response follows (the reviewers comments are in a bold black font, changes in the manuscript are in a red font).
Please see the attachment for full response.

Reviewer 2 Report
This is a safety study, and the authors properly report all the limitations. Still, the case reports and the review of literature in the discussion session are valuable pieces of information, which add knowledge to the field. Obviously, the findings do not back up the conclusion that the treatment “… could significantly reduce tremor in majority of patients who are not candidates for invasive procedures”. As a matter of fact, table 4 reports no changes in tremor in 67% of the subjects at the “last follow up”. Conclusions, therefore, need revision.
As a minor issue, table 5 is probably redundant and it makes it difficult to a reader to grasp the take home message. It would also be of interest to report any possible change of the pharmacological regime following the intervention.
Author Response

(The authors gave the same response as above.)
